# New Fast Acting Glucagon for Recovery from Hypoglycemia, a Life-Threatening Situation: Nasal Powder and Injected Stable Solutions

**DOI:** 10.3390/ijms221910643

**Published:** 2021-09-30

**Authors:** Lucia La Sala, Antonio E. Pontiroli

**Affiliations:** 1IRCCS MultiMedica, Lab of Diabetology and Dysmetabolic Disease, PST Via Fantoli 16/15, 20138 Milan, Italy; 2Dipartimento di Scienze della Salute, Università degli Studi di Milano, 20100 Milan, Italy; antonio.pontiroli@unimi.it

**Keywords:** glucagon, hypoglycemia, diabetes, T1D, T2D, nasal glucagon, dasiglucagon, stable non-aqueous injected glucagon, diabetic complications, glycemic control

## Abstract

The goal of diabetes care is to achieve and maintain good glycemic control over time, so as to prevent or delay the development of micro- and macrovascular complications in type 1 (T1D) and type 2 diabetes (T2D). However, numerous barriers hinder the achievement of this goal, first of all the frequent episodes of hypoglycemia typical in patients treated with insulin as T1D patients, or sulphonylureas as T2D patients. The prevention strategy and treatment of hypoglycemia are important for the well-being of patients with diabetes. Hypoglycemia is strongly associated with an increased risk of cardiovascular disease in diabetic patients, due probably to the release of inflammatory markers and prothrombotic effects triggered by hypoglycemia. Treatment of hypoglycemia is traditionally based on administration of carbohydrates or of glucagon via intramuscular (IM) or subcutaneous injection (SC). The injection of traditional glucagon is cumbersome, such that glucagon is an under-utilized drug. In 1983, it was shown for the first time that intranasal (IN) glucagon increases blood glucose levels in healthy volunteers, and in 1989–1992 that IN glucagon is similar to IM glucagon in resolving hypoglycemia in normal volunteers and in patients with diabetes, both adults and children. IN glucagon was developed in 2010 and continued in 2015; in 2019 IN glucagon obtained approval in the US, Canada, and Europe for severe hypoglycemia in children and adults. In the 2010s, two ready-to-use injectable formulations, a stable non-aqueous glucagon solution and the glucagon analog dasiglucagon, were developed, showing an efficacy similar to traditional glucagon, and approved in the US in 2020 and in 2021, respectively, for severe hypoglycemia in adults and in children. Fast-acting glucagon (nasal administration and injected solutions) appears to represent a major breakthrough in the treatment of severe hypoglycemia in insulin-treated patients with diabetes, both adults and children. It is anticipated that the availability of fast-acting glucagon will expand the use of glucagon, improve overall metabolic control, and prevent hypoglycemia-related complications, in particular cardiovascular complications and cognitive impairment.

## 1. Introduction

### 1.1. Treatment of Patients with Type 1 and Type 2 Diabetes

Today, management of hyperglycemia is possible for nearly all patients with type 1 (T1D) and type 2 (T2D) diabetes mellitus, since the chances of treatment have dramatically changed for both kinds of patients. Oral agents such as metformin are used together with new oral DPP-IV antagonists and SGLT-2 inhibitors, as well as injectable GLP-1 agonists. Insulin analogs with an extremely short or an extremely long duration of action are now in use, and in T2D older drugs such as sulfonylureas are less and less in use, due primarily to their significant risk of hypoglycemia.

### 1.2. Hypoglycemia in Patients with T1D and T2D

Prevention and treatment of hypoglycemia still represents an unmet need; euglycemia without acute metabolic complications still remains a major goal of management in patients with diabetes.

#### 1.2.1. What Is Hypoglycemia, Frequency of Hypoglycemia, and Hypoglycemia Unawareness

Hypoglycemia is defined as a concentration of glucose in the blood that is lower than normal, traditionally defined biochemically as a plasma glucose < 3.5 mmol/L [1]. Hypoglycemia can occur during daytime or night. The main cause of hypoglycemia in diabetes is glucose lowering therapy: exogenous insulin or insulin secretagogues, such as sulfonylureas. Asymptomatic hypoglycemia (plasma glucose < 3.9 mmol/L), symptomatic hypoglycemia (plasma glucose < 3.0 mmol/L), and severe hypoglycemia (SH, plasma glucose not defined, a circumstance where the patient is unconscious and requires the assistance of someone else) are frequent in patients with T1D and T2D who use insulin, or in T2D patients on sulfonylureas [1].

The frequency of hypoglycemia in people with T1D is estimated to be two events per week of non-severe hypoglycemia and one episode of severe hypoglycemia per year [2].

Among insulin-treated patients the frequency of hypoglycemia is greater in T1D than in T2D patients, and depends on intensive insulin treatment, regimens of insulin administration, and age [3,4,5,6,7,8,9,10]. Epidemiology studies have shown that patients with severe hypoglycemia had previous non-severe hypoglycemia diagnoses more frequently than those without [11].

Under normal conditions, a sudden decrease of blood glucose levels is followed by inhibition of insulin release and by sympathetic activation (catecholamines) and hormonal counter-regulation (glucagon, growth hormone and cortisol). Symptoms of hypoglycemia are due to sympathetic activation (sweating, tremor, palpitations, anxiety) and to neuroglycopenia (hunger, slowing cognitive function, sleepiness, seizures, and coma). In people on insulin for long periods, all the responses are usually impaired, and recurrent exposure to hypoglycemia may result in lowering of the plasma glucose concentration, at which point stress responses are initiated, sometimes below the level at which cognitive deterioration starts, leading to impaired awareness of hypoglycemia [12].

Frequent and recurrent hypoglycemia (under 3 mmol/L) in patients with T1D can lead to hypoglycemia unawareness, which puts individuals (moreso adult than pediatric patients) at greater risk of continued and severe hypoglycemia; there is evidence that avoidance of such exposures can restore awareness.

#### 1.2.2. Hypoglycemia as a Life-Threatening Situation

##### Hypoglycemia, Cardiovascular Accidents, and Risk of Death

Acute effects include tachy- and bradycardia, widened arterial pulse pressure and arrhythmias. The arrhythmias have been associated with adrenergic responses and the fall in plasma potassium associated with hypoglycemia. Hypoglycemia produces endothelial dysfunction, an inflammatory response, and creates a coagulopathy, which can last for several days with unknown long-term impacts on cardiovascular health [12,13]. Increased CVD risk is associated with hypoglycemia, especially with serious hypoglycemia events [14]. Symptoms depend on the degree of hypoglycemia [15]; severe hypoglycemia can cause brain damage, and even death. The sudden unexplained death of young people with T1D has been named “Dead in Bed Syndrome” [16,17,18,19,20,21]. Experiments in rats have shown that hypoglycemia increases myocardial susceptibility to Ischemia Reperfusion injury in hearts from animals with and without diabetes. In contrast to hearts from animals without diabetes, the hearts from animals with diabetes are amenable to cardio-protection during hypoglycemia [22].

##### The Link between Hypoglycemia and Cardiovascular Risk

Evidence reporting the link between hypoglycemia and cardiovascular disease has been emerged from large prospective randomized clinical trials [23,24] in which hypoglycemia was identified as the strongest predictor of macrovascular events, adverse clinical outcomes, and mortality in people with T2D. In T1D, the frequency of hypoglycemic events emerging from the Diabetes Control and Complications Trial (DCCT) [6] and the EURODIAB Prospective Complications Study [25] have not demonstrated an increased risk of mortality or fatal cardiovascular disease related to hypoglycemia. In contrast, in T2D the frequency of severe hypoglycemic events is much higher, with intensification of glucose-lowering therapy compared with standard; however, the reported event rate differs between the trials and is not exclusively associated with the HbA_1c_ value, also depending on age, comorbidities, diabetes duration, and treatment regimens. In spite of these differences and concerns, T1D and T2D share the same “common soil” and hypoglycemia seem to induce oxidative stress, inflammation, and endothelial dysfunction in both. Currently there is debate on whether hypoglycemia is an independent risk factor for atherosclerosis [26]. Regarding the cardiovascular effects, the hemodynamic changes associated with hypoglycemia include increased heart rate and systolic blood pressure, reduced peripheral arterial resistance and increased myocardial contractility, stroke volume, and cardiac output [27]. Severe hypoglycemia occurred about two to three times more often in the intensive care unit (ICU), and is more common in people with low cognitive function [28]. In older patients with T2D, dementia is associated with episodes of severe hypoglycemia [29] and these episodes in patients with cardiovascular disease can also lead to myocardial ischaemia and arrhythmia [30]. Both the ACCORD and ADVANCE studies reported higher mortality rates in participants with one or more episodes of severe hypoglycemia, although no cause and effect relationship between these data was established [31,32].

Although the mechanism remains unclear, increased inflammatory cytokines and leukocytosis are reported after hypoglycemia, suggesting a link between hypoglycemia and inflammation [27]. Also, hypoglycemia seems to induce changes in oxidative stress [33] and leads to a disturbance of the physiological redox state as well as to endothelial dysfunction [34]. The fact that hypoglycemia is a defined risk factor for cardiovascular disease in T2D and less so in T1D most likely depends on the different ages of the patients.

##### Cognitive Impairment

In T1D, and in particular T1D-unaware patients, there is a progressive blunting of brain responses in the cortico-striatal and fronto-parietal neurocircuits in response to mild-moderate hypoglycemia [35,36]; these cognitive tasks depend on the hippocampus, which is vulnerable to neuroglycopenia [37]. Hypoglycemia, as an oxidative stressor, may exacerbate chronic hyperglycemia-induced increases in oxidative stress and inflammation, leading to damage to vulnerable brain regions and accelerating cognitive decline [12].

These aspects have been shown in experimental research. It is supposed that prior to glycemic control, hypoglycemia and the degree of rebound hyperglycemia interact synergistically to accelerate oxidative stress and inflammation, which may explain why increased glycemic variability is now increasingly considered a risk factor for the complications of diabetes [38].

#### Quality of Life, Accidents, and Other Impacts

Not only severe, but also symptomatic hypoglycemia negatively affects patient QoL, especially in T2DM; addressing fear of hypoglycemia should be a goal of diabetes education [39]. Falls and trauma have been reported in diabetic patients during hypoglycemia, and a particular caution should be reserved for diabetic patients driving vehicles, exerting themselves, or doing work requiring special attention [40].

In addition, the experience of hypoglycemia leads to fear of hypoglycemia, which in turn can limit optimal glycemic control in many children and adolescents, as well as in adults with T1D. Finally, consideration should be given to societal impacts in the form of loss of driving privileges, restricted employment and campus experiences, high stress between partners, breakdown of family relationships and restricted access to children, as well as economic impacts such as time lost from work as costs beyond the medical costs for healthcare providers) [12].

#### 1.2.3. Prevention and Treatment of Hypoglycemia

Both prevention and treatment of hypoglycemia are crucial to a normal life for patients with diabetes [15]. Prevention is based on education, optimal timing around food or snack ingestion and exercise [41], blood glucose monitoring, and correction of hypoglycemia unawareness [42]. Usage of real time Continuous Glucose Monitoring reduces the number of hypoglycemic events in individuals with T1D treated by multiple daily injected (MDI) and with impaired hypoglycemia awareness or severe hypoglycemia [43]. Treatment of hypoglycemia is still based on administration of carbohydrates (oral or parenteral according to the level of consciousness) or, especially in out-of-hospital environments, of glucagon by intramuscular (IM) or subcutaneous (SC) injection.

### 1.3. The History of Glucagon, Its Use, and Its Limitations

#### 1.3.1. History of Glucagon

In 1923, what was then merely a “component” of the pancreatic extracts was found to induce transient hyperglycemia [44]. Cross-circulation experiments in dogs were of paramount importance to understanding the mutual interactions between alfa and beta cells in the control of blood glucose levels, and to assigning a physiological role to the pancreatic hyperglycemic substance mentioned above [45].

In the 1950s the term glucagon was coined, and in the 1950s, Lilly Research Laboratories were successfully able to extract a pure crystallized form of glucagon [45]; in 1957 the complete amino acid sequence was determined [46].

Glucagon is a 29-amino-acid peptide biologically synthesized by pancreatic alpha cells. It is considered a relevant counterregulatory hormone with a key role in the maintenance of normoglycemia, as it counteracts the effect of insulin in glucose metabolism regulation [47].

#### 1.3.2. Actions of Glucagon

Glucagon mainly exerts its hyperglycemic effects through stimulation of hepatic glycogenolysis. In contrast to the role of the insulin signaling pathway, the glucagon signaling pathway promotes catabolism; through its receptor, glucagon stimulates and releases glucose. The glucagon signaling pathway activates hepatocyte phosphorylase and accelerates glycogenolysis through the cAMP-PK system.

The actions of glucagon are transduced via interaction with a subset of G protein coupled receptors, G_s_α and G_q_. These receptors are linked to the activation of the adenylate cyclase-dependent pathway, which increases intracellular cyclic AMP and activates protein kinase A, and in some cases, stimulation of inositol trisphosphate and calcium influx. Activating G_q_ activates phospholipase C, increases production of inositol 1,4,5-triphosphate, and releases intracellular calcium. Protein kinase A phosphorylates glycogen phosphorylase kinase, which phosphorylates glycogen phosphorylase, which phosphorylates glycogen, causing its breakdown (Figure 1).

In addition, the increased risk of developing T2D could be attributed to some polymorphisms, such as on the human Glucagon receptor (GLU-R) gene, localized to human chromosome 17q25 [48].

#### 1.3.3. Indications of Glucagon

The main indication for glucagon is hypoglycemia in insulin-treated diabetic patients; however, congenital hyperinsulinism and alimentary hypoglycemia in infants are also amenable to glucagon. In contrast to other peptide hormones (e.g., insulin) glucagon does not show a clear dose–response relationship, suggesting that the glycemic response to glucagon is saturable; after IV or IM glucagon administration, there is a clear relationship between doses administered and pharmacokinetic parameters (Cmax, etc.), but progressively increasing doses do not result in dose-related responses of glucose (NDA 020928, Eli Lilly). This has opened the way to the use of glucagon mini-doses that have recently been administered at doses of 100–150 µg instead of 1 mg, both in adults and in infants, to prevent or to treat mild hypoglycemia [49,50,51]. In addition to its effect in contrasting hypoglycemia, glucagon has been used in investigative radiology due to its hypotonic effect on gastric motility [52], and in the diagnosis of various endocrine disorders, i.e., assessment of residual insulin release in diabetes, growth hormone deficiency, and phechromocytoma. Glucagon has cardiovascular effects that can restore heart rate, cardiac output, and blood pressure (inotropic effect) after an overdose of beta-blockers or calcium channel blockers, and in digoxin induced bradycardia and atrioventricular block; glucagon was previously used as an inotropic agent for heart failure, but this indication has been abandoned [53].

#### 1.3.4. Limitations to the Use of Glucagon

Even though it has been available for long time and approved by ADA for resolution of hypoglycemia, injected glucagon has been under-utilized in the US, in Europe, and in Japan. Glucagon is as effective as glucose in the resolution of hypoglycemia [54], but its use is infrequent [55,56].

Glucagon is a peptide hormone and, as such, carries the challenges associated with stability and with the limitations common to administration of most peptides. Reasons for under-utilization are mainly represented by the difficulty of administering glucagon in its traditional form. Glucagon solutions are not stable, owing to the propensity of glucagon to form fibrils once in an aqueous solution [57].

As a result, currently available glucagon emergency kits require reconstitution of lyophilized powder in a diluent immediately prior to IM injection by family members or others who may not be well trained in or comfortable with giving injections (https://myglu.org/polls/1303 accessed on 1 September 2021).

Injection of IM glucagon is so difficult for untrained individuals that some U.S. states allow only trained nurses or health professionals to administer glucagon while children and adolescents with T1D are in school [58]. In a survey conducted by Glu (https://myglu.org/polls/1303 accessed on 1 September 2021; myglu.org, the T1D exchange patient/caregiver online community), approximately 75% of people interviewed declared they rarely or never carried a glucagon emergency kit.

### 1.4. New Fast-Acting Formulations of Glucagon

#### 1.4.1. Nasal Glucagon

The story of nasal glucagon is very long, starting in the 1980s, when a group of investigators demonstrated that IN glucagon drops can raise blood glucose levels in normal subjects [58] and that IN glucagon solutions and IN glucagon powders are similarly effective [59].

IN glucagon without promoters is not absorbed through the nose and is without any effects whatsoever [60,61]. Several promoters have been used by different investigators, such as sodium-glycocholate, 9-lauryl-ether, deoxycholic-acid, didecanoyl-phosphatidylcholine (DDPC) with alpha-cyclodextrin, sodium caprate, and microcrystalline cellulose [59,60,61,62].

During the 1980s, several authors showed that IN glucagon was not different from IM glucagon in resolving hypoglycemia in normal subjects and in adults and children with diabetes [51,63]. In spite of these encouraging results, no pharmaceutical company at that time considered the opportunity to develop intranasal glucagon, likely considering that IM glucagon was effective [54], and that this effectiveness did not justify investment in alternative routes of administration.

Only in 2010 did a Canadian pharmaceutical company (Locemia Solutions, Montreal, QC, Canada) re-examine old data and start development of glucagon for IN administration. The company hypothesized that patients and caregivers might be more comfortable with a needle-free formulation and, on that basis, conducted safety, efficacy, and human factor studies to support registration of the product [64,65,66,67,68].

The project has been continued by Eli Lilly, a U.S. pharmaceutical company that obtained registration for IN glucagon as a powder (with beta-cyclodextrin plus dodecyl-phospho-choline as the promoter) [69,70] for severe hypoglycemia in adult and adolescent insulin users in US, Canada, and Europe in 2019. As shown in Table 1, the recommended dose of IN glucagon (Baqsimi) is 3 mg. The long period from the original finding [62] to pharmaceutical development and final approval of IN glucagon for hypoglycemia is probably one reason why several citations of the pioneering work are absent or confused in the scientific literature.

#### 1.4.2. Glucagon Stable Solution

The glucagon formulation is packaged in a ready-to-deliver auto-injector similar to the epinephrine rescue pens used for anaphylaxis. A freeze-dried powder that contains glucagon along with trehalose, hydroxyethyl, or a similar starch with a glycine buffer (maintaining a pH of 2.0–3.5), a surfactant, an antioxidant, and a chelating agent is combined with a polar aprotic liquid comprising triacetin, thereby preventing the aggregation and fibrillation of the glucagon that is normally propagated by aqueous solutions. This process creates a low-volume, highly-concentrated solution, effective in raising blood glucose levels [71,72,73].

The auto-injector is available in two premeasured doses: a 0.5 mg/0.1 mL dose for pediatric administration and a 1 mg/0.2 mL dose for adolescent and adult administration (Xeris Pharmaceuticals, Inc. Available from www.xerispharma.com/images/xeris-annual-report-2018.pdf, accessed on 1 September 2021). Three studies demonstrated the similarity of this glucagon molecule and of traditional glucagon kits in insulin-induced hypoglycemia in adults and children. Human factors were repeatedly evaluated for the easiness and completeness of administration of this stable solution, also in comparison with traditional glucagon, and found to be superior to traditional glucagon kits [74,75]. The stable solution has proven to be quicker and less complicated to administer by trained and untrained users alike, with a 99% success rate for full dose delivery. The glucagon auto-injector has also been able to more quickly resolve global hypoglycemia symptoms compared to the current traditional glucagon kits (https://scholar.google.com/scholar Christiansen, Cummins, Prestrelski, Junaidi, Poster presented at Diabetes Technology Society’s Diabetes Technology 8–10 November 2018, Bethesda).

#### 1.4.3. Dasiglucagon

The glucagon analog dasiglucagon is the first glucagon product to be provided as an aqueous formulation, ready-to use. Similar to endogenous glucagon, dasiglucagon is composed of 29 amino acids, and carries seven amino acid substitutions to increase the physical and chemical stability in aqueous media. The aqueous formulation does not require reconstitution before injection.

Dasiglucagon has specificity for the glucagon receptor and potency comparable to that of native glucagon. In T1D patients, dasiglucagon restored euglycemia after insulin-induced hypoglycemia at doses from 0.1 to 1.0 mg [76], and provided rapid and effective reversal of hypoglycemia in adults with T1D, with safety and tolerability similar to those reported for reconstituted glucagon injection [77,78,79]. Dasiglucagon was approved in the US in 2021 [80].

Table 1 summarizes the main characteristics of the three ready-to-use glucagon formulations approved by regulatory agencies so far for severe hypoglycemia, and of the traditional glucagon emergency kits. The dosages of the three formulations are different, likely due to differences in the molecules or the delivery route. Therefore, maximum concentrations in plasma glucagon levels differ, while half-lives are similar. Time to peak glucagon concentrations and to blood glucose rise are different, but the degree of blood glucose rise and the duration of increased blood glucose levels are similar. For all formulations, a second dose is recommended in the case of failure. Side effects are similar for the three formulations, based on the nature of the molecule; in addition, side effects due to local administration (head and facial discomfort, increased lacrimation, nasal discomfort) appear with nasal glucagon.

For all formulations, real world situations were evaluated to understand the usability of the new drugs. In particular, studies were based on comparisons between the new formulations and the traditional emergency glucagon kits, to assess easiness and therefore completeness of administration. For all three drugs, positive results were obtained, showing that caregivers did not have to be familiar with these drugs, in contrast to traditional emergency glucagon kits. For nasal glucagon, additional studies evaluated drug administration under situations of common cold and of use of nasal decongestants, showing that the effectiveness of nasal glucagon was not affected by these situations. An interesting difference between nasal glucagon and the two other formulations is the time employed for development. For nasal glucagon it took from 1983 to 2019 to be developed and approved; for the other two drugs it took approximately six years. At present, in the absence of comparative studies, all three formulations appear valid for emergency treatment of severe hypoglycemia. A common aspect is that the cost-effectiveness ratio is favorable for fast-acting glucagon compared to traditional glucagon preparations [81].

### 1.5. The Future

#### 1.5.1. The Need for Better Education and Awareness of Hypoglycemia

Severe hypoglycemia still represents a serious problem in children, even though new insulin analogs, better insulin administration regimes, and close glucose monitoring have reduced its frequency [82]. In contrast, between the age 18–44 the frequency is still high, and does not show the same declining [83]. Since by definition severe hypoglycemia requires the help of a third person as caregivers, education of such potential caregivers as parents, teachers, friends and colleagues, medical students and nurses is of paramount importance; for instance, training in how to recognize hypoglycemia as well as ad hoc educational programs should be implemented [84,85,86]. In addition, adequate intervention requires education, especially since glucagon represents the mainstay of hypoglycemia treatment of in out-of-hospital environments. For instance, in a study where trained caregivers administered glucagon to treat an episode of simulated SH, injectable glucagon was properly administered by only 2 of 16 caregivers, (average time, 1.89 min), while IN glucagon was successfully administered by 15 of 16 caregivers (average time, 16 s) [66]. Finally, patients with diabetes should be made aware of the existence of fast-acting glucagon. A telephone survey conducted in 1997 ascertained opinions on the currently available glucagon emergency kits among patients with T1D, 67% of whom stated that they would prefer IN-administered glucagon if and when available, and 82% of whom believed that family members, teachers and colleagues would prefer to administer emergency therapy via IN in cases of severe hypoglycemia. In a survey performed to assess awareness of new fast-acting glucagon formulations, 797 patients and parents responded. The majority knew about nasal glucagon (530/797, 66.5%), but only 41/797 (5.1%) had bought it, and children and adolescents aged ≥6 years were significantly more likely to have heard of nasal glucagon than the parents of younger children [87]. Therefore, only a few pediatric patients know about nasal glucagon, and consequently still prefer the injectable form.

#### 1.5.2. Expanding the Use of Glucagon

We can foresee that because of ease of administration of the three new formulations the use of glucagon to resolve severe hypoglycemia will likely expand dramatically. This will open the path to innovative uses of nasal glucagon, of self-injected stable glucagon, and of dasiglucagon. Of course, such increased use might also reveal other as yet unknown metabolic effects and side effects. For instance, even though not approved, one might think of their use as a self-remedy, or with the aid of a caregiver, to prevent severe hypoglycemia when under mild symptomatic hypoglycemia; fully randomized studies are necessary to evaluate this possibility. In addition, the use of these drugs in other types of hypoglycemia such as congenital hyperinsulinism, post-bariatric surgery hypoglycemia, alimentary hypoglycemia, and exercise-induced hypoglycemia, deserves greater attention.

#### 1.5.3. Unconscious Patients

With the exception of one study dealing with spontaneous hypoglycemia [63], the majority of studies were based on experimental and intentional insulin-induced hypoglycemia. This may not truly represent real-world use of the product, as patients in the trials received high doses of insulin to achieve hypoglycemia; this might have limited their ability to see a rise in glucose or determine effectiveness. In addition, no study specifically compared IN and IM/SC glucagon in unconscious patients; however, in the paper by Seaquist et al. [67] there were 12 episodes of severe hypoglycemia, that is, with unconsciousness and/or convulsions, that were rapidly and successfully treated by the patient’s caregivers using IN glucagon. Moreover, both in that paper and in the paper by Deeb et al. [81], caregivers had the option to administer IM glucagon if they felt the patient was not responding to IN glucagon, and not a single participant had to receive IM glucagon.

#### 1.5.4. Improvement of Metabolic Control and Prevention of Complications

As said before, an unmet need is still represented by prevention and treatment of hypoglycemia; euglycemia without acute metabolic complications still remains a major goal of management in patients with diabetes. Even though nasal glucagon, self-injected stable glucagon, and dasiglucagon are effective in resolving severe hypoglycemia, it would be reassuring to show that overall metabolic control improves as well in patients using these emergency drugs. Moreover, it would be reassuring to show that use of these drugs helps to prevent hypoglycemia-related complications such as cardiovascular disease and cognitive impairment. Long term studies are required to show these possible benefits.

#### 1.5.5. Development of New Glucagon Molecules

Glucagon still represents the fastest and most effective remedy to hypoglycemia, provided it is used by a trained person. The complexity of preparing glucagon for administration is the main reason why alternative routes of administration and other formulations have been actively searched for. In addition to the three drugs approved, other approaches to effective glucagon delivery have been attempted; some of them are still under development, while other routes have been abandoned. For instance, Biochaperone Glucagon is made of polymers, oligomers, and organic compounds that can form a complex with glucagon and improve its stability in aqueous solution. BIOD-961 is a lyophilized glucagon formulation that, while it does require reconstitution, has an auto-reconstitution device which has been developed. Another glucagon agonist, SAR438544, was developed for subcutaneous delivery; however, this medication is no longer under development [88,89,90]. The oral route has been examined for glucagon and has too many counter-indications to be a viable option for emergency glucagon administration; swallowing is a requirement for any oral medication and is impossible during a hypoglycemic emergency [89]. A transdermal approach (ZP Glucagon) has been developed for delivering glucagon, and a safety and efficacy study has been completed, but development has not been continued [88,89,90].

## 2. Conclusions

Glucagon remains the recommended remedy for hypoglycemia in out-of-hospital environments; however, its use is less frequent than expected due to the complexity of administration. The recent development of alternative routes of administration and of new fast-acting stable solutions for injection raises the necessity of making glucagon able to be easily administered by untrained persons, whether patients or caregivers. It can be anticipated that doing this will lead to expansion of the use of glucagon in patients with hypoglycemia, thus making possible the attainment of euglycemia without hypoglycemia. At present, in the absence of comparative head-to-head studies, all of the new formulations appear to be valid remedies; however, patients’ expectations are in favor of nasal administration [78].

## Figures and Tables

**Figure 1 ijms-22-10643-f001:**
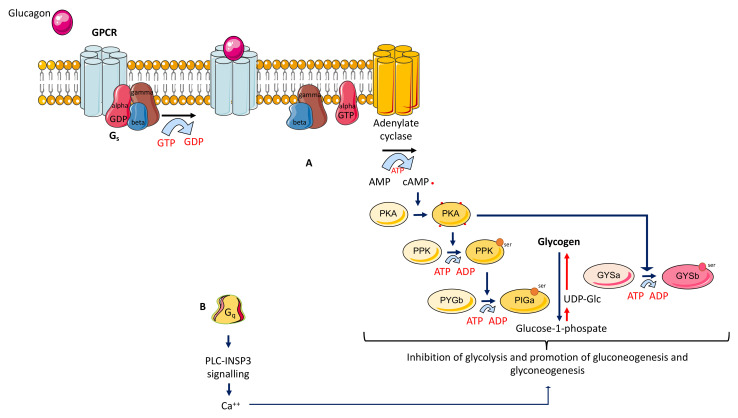
Depiction of glucagon signalling. The glucagon signaling pathway promotes glycogenolysis and gluconeogenesis, resulting in a significant increase in blood glucose. Glucagon (red ball) binds to G protein-coupled receptor (GPCR) and provides a conformational status that activates G proteins. (**A**) The heterotrimeric G-proteins with α, β, and γ subunits, interact with the receptor, and replace the GDP molecule (to the α subunit) with a GTP molecule. Thus, α subunit leaves the β and γ subunits, and specifically activates adenylate cyclase. The latter catalyzes the conversion of ATP to cAMP (cyclic Adenosine Mono Phosphate), which binds to protein kinase A (PKA). PKA, a cAMP-dependent protein kinase, phosphorylate a single serine residue of the bifunctional polypeptide chain containing both the enzymes fructose 2,6-bisphosphatase and phosphofructokinase-2. This regulates the reaction catalyzing fructose 2,6-bisphosphate (a potent activator of phosphofructokinase-1, the enzyme that is the primary regulatory step of glycolysis) by slowing the rate of its formation, thereby inhibiting the flux of the glycolysis pathway and allowing gluconeogenesis to predominate. The inactive form of glycogen phosphorylase b (PYG b) undergoes phosphorylation, allowing it to convert into the active form, phosphorylase a (PYG a), which releases glucose 1-phosphate from glycogen polymers. (**B**) The binding with a subunit Gq leads to activation of the phospholipase-C (PLC)-inositol 3 phosphate (I3P) pathway, inducing the release of Ca^++^.

**Table 1 ijms-22-10643-t001:** New ready-to-use glucagon preparations approved for treatment of severe hypoglycemia, compared to traditional emergency kits.

Preparation	Intranasal Glucagon	Subcutaneous Glucagon	Subcutaneous Dasiglucagon	Traditional Intramuscular/Subcutaneous Glucagon Emergency Kits
Molecule	glucagon	glucagon	7 substitutions in glucagon	glucagon
Formulation	Dry powder	Stable non acqueous solution	Stable acqueous solution	Acqueous solution to be reconstituted before use
Dose suggested adults	3 mg	1 mg	0.6 mg	1 mg
Dose suggested infants	3 mg	0.5 mg *	0.6 mg **	0.5 mg ***
Pharmacokinetics (PK) §				
c-max	3155 pg/mL	2481 pg/mL	1570 pmol/L (5444 pg/mL) #	6900 pg/mL
t-max	15 min	50 min	35 min	13 min
T ½	38 min	32 min	30 min	26 min
Pharmacodynamics (PD) §				
Time to BG increase °°	5 min	9 min	6–10 min	6–10 min
Increase of BG °°	+102–140 mg/dL	+123–145 mg/dL	+120–140 mg/dL	+132–138 mg/dL
-Duration of BG increase °°	>90 min	>90 min	>90 min	>90 min
Repeat dose in the case of failure during the first 15 min	+	+	+	+
Real world studies	Performed	Performed	Performed	Not performed
Side effects	Nausea and vomiting, head and facial discomfort, increased lacrimation, nasal discomfort	Nausea and vomiting, headache	Nausea and vomiting, headache	Nausea and vomiting, headache
Contraindications	As for glucagon §§	As for glucagon §§	As for glucagon §§	§§
Approval and date	FDA, 2019 EMA, 2019	FDA, 2020 (EMA, application 2021)	FDA, 2021	FDA 1998, EMA 1991 °°°
Approved indication	Severe hypoglycemia	Severe hypoglycemia	Severe hypoglycemia	Severe hypoglycemia
Commercial name	Baqsimi	Gvoke Hypopen, (Ogluo)	Zegalogue	Glucagon GlucaGen HypoKit

* recommended for age > 2 years and with weight < 45 kg; ** recommended for age > 6 years; *** recommended for children weighing < 20 kg; § PK and PD for the higher dose; # calculated by the authors on the basis of conversion to SI units; °° BG = blood glucose; §§ allergy to glucagon, suspected neoplasia of the pancreas (insulinoma or glucagonoma), of the adrenals (pheochromocytoma); °°° the two rDNA-origin drugs (Glucagon, GlucaGen HypoKit) were approved in the 1990′s by FDA and EMA; the extractive drugs had been approved in the 1960′s.

## Data Availability

Not applicable.

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
