# Peer review of "New Fast Acting Glucagon for Recovery from Hypoglycemia, a Life-Threatening Situation: Nasal Powder and Injected Stable Solutions"

_ijms, 2021, doi:10.3390/ijms221910643_

Round 1

Reviewer 1 Report

The manuscript focused on a brief introduction centered on the application of glucagon in dealing with hyoglycemia. The content is divided into five parts: treatment of patients with T1DM and T2DM; Hypoglycemia in patients with T1DM and T2DM; The history of glucgon, its use and its limitaions; new fast-acting formulations of glucagon; and the future.

Suggestions for the improvement.

  1. The contents in parts are too lengthy. A concise content is better for readers’ reading.
  2. There are several etiologies and situiations of hypoglycemia needing glucagon treatment other than T1DM and T2DM. To expand the utility of glucagon in dealing with hypoglycemia, the applicable diseases and situiations should be included.
  3. In addition to category introduction, a list and brief introduction of new candidates in each category are helpful.

Author Response

Referee 1

The manuscript focused on a brief introduction centered on the application of glucagon in dealing with hyoglycemia. The content is divided into five parts: treatment of patients with T1DM and T2DM; Hypoglycemia in patients with T1DM and T2DM; The history of glucagon, its use and its limitaions; new fast-acting formulations of glucagon; and the future.

Suggestions for the improvement.

  1. The contents in parts are too lengthy. A concise content is better for readers’ reading.

Answer

Thank you for the criticism. We have thoroughly revised the MS; several paragraphs have been shortened, in spite of new paragraphs added (see point 2.); namely, sections 1.2.1., 1.2.2.1, 1.2.2.3, 1.3.1, 1.4.1, 1.5.1

  1. There are several etiologies and situations of hypoglycemia needing glucagon treatment other than T1DM and T2DM. To expand the utility of glucagon in dealing with hypoglycemia, the applicable diseases and situations should be included.

Answer

Thank you for the suggestion. A section has been included 1.3.3. to talk on other effects and uses of  glucagon

  1. In addition to category introduction, a list and brief introduction of new candidates in each category are helpful.

Answer

Thank you for your criticism. A brief sections (1.5.5.)has been added to mention the studies and the alternative routes of administration still ongoing or discontinued

Reviewer 2 Report

This manuscript compares three different formulations of glucagon as a treatment for hypoglycaemia associated with diabetes.

The manuscript has a number of shortcomings:

  1. The title should really convey the purpose of the study. Therefore, I suggest that it should be modified to, for example: A comparison of three new fast acting formulations of glucagon for recovery from a life-threatening hypoglycemia. situation. Nasal powder and injected stable solutions
  2. Table 1 is good, but any benefit over standard preparations of glucagon is not clear. Therefore, this comparison should be added.
  3. A graph comparing the PK profile of the different formulations would be helpful, especially if it contained pharmacodynamic data.
  4. ‘BG’ needs to be defined.
  5. A graph showing PK-PD relationships is essential and so needs to be added.
  6. What is the ‘n’ for the various groups in the Table?
  7. Section 1.4.1: Eli Lilly, a U.S. pharmaceutical ‘company’ – not ‘industry’
  8. A paragraph entitled ‘Conclusions’ would be helpful at the end of the manuscript.
  9. Which formulation do the authors consider to be best ?

Author Response

This manuscript compares three different formulations of glucagon as a treatment fo hypoglycaemia  associated with diabetes.

 The manuscript has a number of shortcomings:

  1. The title should really convey the purpose of the study. Therefore, I suggest that it should be modified to, for example: A comparison of three new fast acting formulations of glucagon for recovery from a life-threatening hypoglycemia. situation. Nasal powder and injected stable solutions

Answer

Thank you for the suggestion. The aim of this review was to describe newapproved glucagon preparations, not to compare them. There is no possibility to compare the new preparations, because there is no direct comparison in the scientific literature [except for ref. 78 on patients perspectives and expectations]

Table 1 is good, but any benefit over standard preparations of glucagon is not clear. Therefore, this comparison should be added.

Answer

Thank you for the criticism a column has been added for traditional glucagon kits. As said in the text and in the conclusions, the new preparations have been developed to tackle the complexity of administration of traditional glucagon, not to improve its efficacy.

  1. A graph comparing the PK profile of the different formulations would be helpful, especially if it contained pharmacodynamic data.

Answer

Thank you for the suggestion. However, aside from what is shown in Table 1, there are no direct comparisons in the scientific literature; such a graph would be arbitrarily drawn by the authors of this review

  1. ‘BG’ needs to be defined.

Answer

Thank you. Done as required

  1. A graph showing PK-PD relationships is essential and so needs to be added.

Answer

Thank you for the suggestion. However, aside from what is shown in Table 1, there are no direct comparison in the scientific literature; such a graph would be arbitrarily drawn by the authors of this review

  1. What is the ‘n’ for the various groups in the Table?

Answer

The numbers are now accompanied by the units of measure

  1. Section 1.4.1: Eli Lilly, a U.S. pharmaceutical ‘company’ – not ‘industry’

Answer

Thank you. Done as suggested

  1. A paragraph entitled ‘Conclusions’ would be helpful at the end of the manuscript.

Answer

Thank you for the suggestion. This has been included

  1. Which formulation do the authors consider to be best ?

Answer

Thank you for the suggestion. As we now write in the conclusions, “At present, in the absence of comparative head-to-head study, all new formulations appear valid remedies; only patients expectations are in favor of nasal administration [78].”

Round 2

Reviewer 1 Report

There was no additional comment.